NFIB promotes the migration and progression of kidney renal clear cell carcinoma by regulating PINK1 transcription

Wang Ninghua wnh120150930@163.com
Yuan Jing
Liu Fei
Wei Jun
Liu Yu
Xue Mei
Dong Rui
1 Department of Urology, Hanyang Hospital , Wuhan , Hubei , China
Verghese Shilpi
Electronic publication date: 2021 Apr 23
Publication date: 2021
Volume: 9
Electronic Location ID: e10848
Received 2020 Aug 11; Accepted 2021 Jan 6
Copyright: ©2021 Wang et al.
Copyright year: 2021
Copyright holder: Wang et al.
License: This is an open access article distributed under the terms of the Creative Commons Attribution License, which permits unrestricted use, distribution, reproduction and adaptation in any medium and for any purpose provided that it is properly attributed. For attribution, the original author(s), title, publication source (PeerJ) and either DOI or URL of the article must be cited.
License URL: https://creativecommons.org/licenses/by/4.0/

Keywords: NFIB, PINK1, Kidney renal clear cell carcinoma, Metastasis, Progression

Funding: The authors received no funding for this work.

==============================
Kidney renal clear cell carcinoma (KIRC) is the most common and aggressive type of renal cell carcinoma. Due to high mortality rate, high metastasis rate and chemical resistance, the prognosis of KIRC patients is poor. Therefore, it is necessary to study the mechanisms of KIRC development and to develop more effective prognostic molecular biomarkers to help clinical patients. In our study, we used The Cancer Genome Atlas (TCGA) and Gene Expression Omnibus (GEO) databases to investigate that the expression of nuclear factor I B (NFIB) is significantly higher in KIRC than in adjacent tissues. Moreover, NFIB expression levels are associated with multiple clinical pathological parameters of KIRC, and KIRC patients with high NFIB expression have poor prognosis, suggesting that NFIB may play vital roles in the malignant development of KIRC. Further studies demonstrated that NFIB could promote the progression and metastasis of KIRC and participate in the regulation of PTEN induced kinase 1 (PINK1). Furthermore, we used chromatin immunoprecipitation (ChIP) experiments to confirm that NFIB binds to the PINK1 promoter and regulates its expression at the transcriptional level. Further experiments also confirmed the important roles of PINK1 in promoting the development of tumors by NFIB. Hence, our data provide a new NFIB-mediated regulatory mechanism for the tumor progression of KIRC and suggest that NFIB can be applied as a new predictor and therapeutic target for KIRC.

Introduction

Renal cell carcinoma (RCC) is one of the most common primary malignant tumors of the kidney, among which kidney renal clear cell carcinoma (KIRC) is the most common and aggressive type, accounting for approximately 60%–70% of RCC histological subtypes (Rini, Campbell & Escudier, 2009; Schmidinger Daniel, James & Vincenzo, 2017). An estimated 63,990 new cases and 14,400 deaths (including 4940 women and 9470 men) were caused by KIRC in 2017 worldwide (Pang et al., 2018). Due to high mortality rate, high metastasis rate and chemical resistance, the prognosis of KIRC patients is poor (Chaffer & Weinberg, 2011; Kim et al., 2011). Genomic studies have identified several genes involved in the progression of KIRC, including BRCA1-associated protein-1 (BAP1), polybromo 1 (PBRM1), SET domain 2 (SETD2) and von-Hippel Lindau tumor suppressor (VHL) (Piva et al., 2015; Nargund et al., 2017). Patients with KIRC are usually treated with standard surgical resection. However, recent studies have shown that the quality of life of KIRC patients varies greatly, and 30% of local tumor patients eventually develop metastases (Hutson & Figlin, 2007). Tumor metastasis results in 90% of cancer-related deaths and poor outcomes for all cancer patients, including KIRC (Moch et al., 2016). More unfortunately, after receiving medical treatment, most KIRC patients eventually develop progressive disease due to intrinsic resistance or acquired resistance (Hsieh et al., 2017; Bergers & Hanahan, 2008). Therefore, it is necessary to study the mechanisms of KIRC development and to develop more effective prognostic molecular biomarkers to help clinical patients.

Site-specific DNA binding proteins in the nuclear factor I (NFI) family play an important role in DNA replication and in the regulation of transcription of a variety of cellular genes (Gronostajski, 2000). The family consists of four genes (NFIA, NFIB, NFIC and NFIX) encoding proteins that interact with DNA as homologous dimers or heterodimers (Kruse & Sippel, 1994). Transcriptome analysis has shown that NFIB is generally more highly expressed in human tumor tissues, which supports the important role of NFIB in tumor biological processes (GTEx Consortium, 2015; Wu et al. 2017) noted that NFIB is associated with key molecular events that drive nonsmall cell lung cancer metastasis. In addition, studies have shown that NFIB directly promotes EZH2 expression and mediates highly invasive and migratory phenotypes in melanoma (Fane et al., 2017). Although NFIB has an established role as an oncogene in nonsmall cell lung cancer and melanoma, it has also been reported that NFIB has a tumor suppressor function in some cancer types. Vo TM et al. demonstrated that NFIB exhibits tumor suppressor activity in glioblastoma, the expression level of NFIB was inversely correlated with astrocytoma tumor grade, and the ectopic expression of NFIB significantly inhibited tumor growth in vivo (Vo et al., 2019). However, to date, no previous studies have reported the roles of NFIB in the malignant development of KIRC and its mechanisms in the tumor growth and metastasis of KIRC.

In our study, we found that NFIB was highly expressed in KIRC through The Cancer Genome Atlas (TCGA) and Gene Expression Omnibus (GEO) data analysis. To further validate our results, we used cell and clinical samples for repeated validation. In vitro, we found that NFIB could significantly promote the proliferation and migration of KIRC cells, which has greatly attracted our interest. Using bioinformatics analysis, we found that PTEN-induced putative kinase 1 (PINK1) may be a downstream site of NFIB; thus, we further demonstrated the promotion roles of PINK1 in NFIB expression and found a binding site for NFIB on the PINK1 promoter using a ChIP assay. In conclusion, we confirmed that NFIB can promote the proliferation and metastasis of tumor cells by transcriptionally regulating the expression of PINK1 in KIRC.

Materials & methods

Patients and clinical samples

Thirty pairs of clinical tissue samples of KIRC were collected from patients undergoing surgery at the Department of Urology, Hanyang Hospital (Wuhan, China) from September 2015 to December 2015. The diagnosis of KIRC was confirmed by the original histopathological reports. The tissue samples were rapidly fixed in formalin for paraffin embedding. All clinical samples were collected with written informed consent to participate in this study. The study was approved by the Ethics Committee of the Academic Medical Center of Wuhan University of Science and Technology (No. 2020-IEC1453).

Cell culture

The source of the tubular epithelial cell line (HKC-5; BNCC100598) and KIRC cell lines (LoMet-ccRCC and 786-0) were obtained from the BeNa Culture Collection (Beijing, China). Cell culture methods were performed as previously described (Yuan et al., 2019) .

Transfection

The siRNA and normal control (NC) siRNAs for NFIB and PINK1 were purchased from RiboBio (Guangdong, China). The sequences of siRNAs are as follows: NFIB sense: 5′-CCAGGUGGUGAAGAAUCUATT-3′, antisense: 5′-UAGAUUCUUCACCACCUGGTT-3′; PINK1 sense: 5′- CAAGACUUGGAGCUAAAAATT-3′, antisense: 5′- UUUUUAGCUC CAAGUCUUGTT-3′. The overexpression plasmids of NFIB and PINK1 were constructed and cloned into the CV061 vector (GeneChem, Shanghai, China) between the EcoRI and HindIII sites. Once the cells reached 30–40% confluence, LoMet-ccRCC and 786-0 cells were transfected using Lipofectamine 3000 reagent (Invitrogen, MA, USA) according to the manufacturer’s instructions. Subsequent treatments were performed after the cells were cultured for an additional 48 hours.

CRISPR-Cas9

NFIB-knockout cells were generated using CRISPR-Cas9 gene editing, and the sgRNA targeting NFIB gene was cloned into GV371-U6-NFIB sgRNA-SV40-EGFP and GV371-CMV-hSpCas9-SV40-Puro, respectively. The sgRNA sequence is as follows: GGGAGTGTCTCCTGCCGAGA. In addition, LV-sgRNA and LV-Cas9 lentiviruses were also produced (GeneChem, Shanghai, China). LV-Cas9 was seeded in 786-0 cells with a MOI of 2 after 3 days of puromycin selection with a final concentration of 5 µg/ml. LV-sgRNA was then seeded, and another 3 days later, the cells were harvested for further detection.

qRT-PCR

Quantitative real-time reverse transcription polymerase chain reaction (qRT-PCR) was performed as previously described (Yuan et al., 2019). The following primers were used: NFIB forward, 5′-GTAATTAGAGCTGTGACA-3′; NFIB reverse, 5 ′-TGTATGCCATCTAGTGGAT-3′; PINK1 forward, 5′-CGTGGCGAGCTGTAATTGAC-3′; PINK1 reverse, 5′-GGCTCCGCAGACGTTATCTA-3 ′; GAPDH forward, 5′-GAAGGTGAAGGTCGGAGTC-3′; and GAPDH reverse, 5′-GAAGATGGTGATGGGATT-3′. All reactions were performed in triplicate. The relative expression levels of the genes were calculated using the 2−ΔΔCq method.

Western blotting

Total protein lysates from the cultured cells were extracted using a lysis buffer containing proteinase inhibitors (Sigma-Aldrich, Darmstadt, Germany). Proteins were denatured, separated in 10% polyacrylamide gels, and transferred to polyvinylidene fluoride (PVDF) membranes for probing with the following primary antibodies: anti-NFIB (1:500, Abcam, ab186738, Cambridge, UK), PINK1 (1:500, Abcam, ab216144, Cambridge, UK) and anti-GAPDH (1:2000, Proteintech, 10494-1-AP, Wuhan, China). For detection, anti-rabbit secondary antibodies conjugated to horseradish peroxidase (1:3000, Proteintech, Wuhan, China) were used. Band signals were visualized by an enhanced chemiluminescence detection kit (Meilunbio, Dalian, China). All western blotting was performed in triplicate.

Cell proliferation assay

Cells were seeded into 96-well plates at a density of 3 × 103 cells per well and observed for 1 to 5 days. Then, 20 µl of MTT (5 mg/mL) was added to each well, and the media were replaced with 200 mL of DMSO (Sigma) after incubation for 4 h. When the crystals were completely dissolved, a microplate reader was used to determine the resulting absorbance at 570 nm. All MTT assays were performed in triplicate.

Colony formation assay

Cells of treatments and experimental groups were respectively seeded into six-well plates at a density of 500 cells per well and cultured with complete medium at 37 °C in 5% CO2 for 15 days. The colonies were then stained with 1% crystal violet (Sigma-Aldrich, Darmstadt, Germany) for 30 min and counted. All experiments were performed in triplicate.

Migration and wound-healing assay

Transwell chambers (Beaverbio, Jiangsu, China) were used to assess KIRC cell metastasis ability. Suspensions of 8 × 104 cells in 200 µl of FBS-free medium (Sigma-Aldrich, Darmstadt, Germany) were added to the top chamber, and medium containing 20% FBS (Sciencell, CA, USA) was added to the bottom chamber to serve as the chemotaxin. The cells were cultivated at 37 °C in 5% CO2 for 48 h, and the cells that failed to migrate were wiped off the top of the membranes. The migrated cells attached to the bottom chamber were fixed and stained. Cells from five random fields were quantified by light microscopy. To determine the invasion ability of KIRC cells, invasion assay was performed by transwell migration chambers coated with Matrigel (Sigma).

For the wound-healing assay, the cells were seeded into 12-well plates (Beaverbio, Jiangsu, China) and grown to 80–90% confluence at 37 °C in 5% CO2. The cell monolayers were prepared with a 200 µl pipette tip and cultured in serum-free medium. Cell migration was assessed by microscopy at 0 and 48 h and objectively analyzed with ImageJ 1.8.0. All experiments were performed in triplicate.

Immunohistochemical

Paraffin-embedded clinical tissues were cut into 5- µm-thick sections and placed on glass slides. The tissue sections were deparaffinized and subjected to antigen retrieval using 0.01 m citric acid buffer (pH 6.0) for 15 min and incubated overnight at 4 °C with primary antibodies against NFIB (1:200, Abcam, ab186738, Cambridge, UK). After three washes with Tris-buffered saline, the sections were incubated with a horseradish peroxidase-conjugated secondary antibody (1:100, Boster) for 1 hour at room temperature. The immunohistochemical (IHC) staining results were evaluated by two independent observers. When staining on the nucleus and ≥30% positive cells in the section, immunohistochemical staining of NFIB was estimated to be positive.

Chromatin immunoprecipitation

Chromatin immunoprecipitation (ChIP) assays were performed using the SimpleChIP Enzymatic Chromatin IP Kit (CST, #9005, MA, USA) according to the manufacturer’s recommended protocol. Cells were crosslinked with formaldehyde and sonicated to an average size of 150 to 900 bp. Then, chromatin extracts were immunoprecipitated using 4 µg of monoclonal anti-NFIB antibody (Abcam, ab186738, Cambridge, UK). The purified DNA was subjected to qPCR to amplify the binding sites of the PINK1 promoter region. All ChIP assays were performed in triplicate.

Data acquisition and statistical analysis

Raw KIRC data containing mRNA sequencing and clinical information were obtained from the TCGA Genome Data Analysis Center (http://gdac.broadinstitute.org/runs/analyses__latest/reports/cancer/STAD/). In addition, the KIRC mRNA expression array of GSE83999 and GSE53757 were acquired from the NCBI GEO (Perron et al., 2018; Von Roemeling et al., 2014). The data analysis methods were performed as previously described (Yuan et al., 2019). The R ‘limma’ Bioconductor package was used to screen the differentially expressed genes (DEGs) between KIRC and adjacent tissues based on the following criteria: Fold change (FC), —log2(FC)—>1; and false discovery rate (FDR) <0.05. Adjusted P < 0.05 was used to define a gene as a DEG. Database for Annotation, Visualization and Integrated Discovery (DAVID) v6.8 (david-d. ncifcrf.gov/) was used to analyze functional enrichment among DEGs. In addition, only those Kyoto Encyclopedia of Genes and Genomes (KEGG) pathways with P ≤ 0.05 and ≥10 enriched genes were considered significant. The 2000bp sequence upstream of the PINK1 gene was used as the gene promoter sequence. The open database JASPAR (http://jaspar.genereg.net) and PROMO (http://alggen.lsi.upc.es/cgi-bin/ promo_v3/promo/promoinit.cgi?dirDB=TF_8.3) were used to predicate the potential NFIB sites on the PINK1 gene promoter.

The assay data were analyzed by SPSS 13.0 and presented as the mean value ± standard deviation (SD). Depending on the type of experiment performed, the assay results were analyzed through two-tailed unpaired or paired Student’s t test. The relationships between the expression of NFIB and the clinical pathological characteristics of KIRC were analyzed by chi square (X2) test. The statistical significances between the groups are presented as P values. When p < 0.05, the results were considered statistically significant.

Figure 1 NFIB was highly expressed and was associated with poor prognosis in KIRC.

(A) Venn plot diagram of differentially expressed genes in TCGA and GEO (GSE83999 and GSE53757) data sets. (B) Top10 mRNA expression profiles (FPKM) and identification of NFIB as differentially expressed between KIRC and normal tissues based on the TCGA data set. (C) Volcano plot of differentially expressed mRNAs between KIRC and normal tissues based on the TCGA data set. Up, upregulation; Down, downregulation. The levels of NFIB were significantly decreased in KIRC (LoMet-ccRCC, 786-0) compared with the human renal tubular epithelial cell line HKC-5 as indicated by (D) qRT-PCR and (E) western blot assay respectively. The high expression of NFIB in KIRC tissues compared with matched adjacent noncancerous tissues via (F) IHC (Scale bars = 100 µm) and (G) qRT-PCR. Kaplan–Meier curves of KIRC patients stratified based on the expression of NFIB in KIRC tissues (high or low) based on (H) TCGA data set, (I) the online data sites GEPIA (http://gepia.cancer-pku.cn). The data D were presented as means ± SD of three independent experiments. Values are significant at * P < 0.05, as demonstrated by paired Student’s t test.

Results

NFIB is overexpressed in human KIRC

To identify novel genes that potentially drive KIRC tumorigenesis, we performed differential expression analysis using the dataset from TCGA (Table S1) and the datasets from GEO (Fig. 1A and Tables S2–S3). Hierarchical cluster analysis revealed alterations in the mRNA transcript expression between KIRC and paired noncancerous kidney (NK) tissue among the three datasets (Fig. 1B). NFIB was one of the prominently upregulated mRNAs in KIRC tissues compared with NK tissues (Fig. 1B and 1C). After a literature search, we found that the roles of NFIB in the development of KIRC have not been reported, so we aimed to further investigate its roles in driving KIRC tumorigenesis. To confirm these results, we further tested the expression levels of NFIB in cell lines. Compared with normal human renal tubular epithelial cell lines, KIRC cell lines showed significantly increased mRNA expression levels of NFIB (Fig. 1D), and the western blot results were consistent with the results obtained from RT-qPCR (Fig. 1E). In addition, NFIB expression was detected in 30 paired KIRC and NK clinical samples using IHC analysis. The expression level of NFIB was significantly higher in KIRC tissues than in paired NK clinical tissues (Fig. 1F and 1G). Taken together, these results demonstrate a novel dysregulated gene, NFIB, in KIRC.

NFIB is associated with KIRC progression and worse prognosis

To explore the clinical significance of NFIB, we assessed the correlation between the expression levels of NFIB and the individual clinicopathological features of KIRC patients using the dataset from TCGA (Table S4). As shown in Table 1, the results indicated that the overexpression of NFIB was significantly associated with poor histological grade (P = 0.008), pathological stage (P = 0.011), T stage (P = 0.035), lymphatic invasion (P < 0.001) and distant metastasis (P = 0.019). Moreover, we found that high NFIB expression was significantly associated with worse prognosis in patients (Fig. 1H, HR: 0.175, 95% CI: [0.091, 0.336]), and we obtained consistent results (Fig. 1I) with the online data sites GEPIA (http://gepia.cancer-pku.cn) (Tang et al., 2017).

Table 1 Association of clinicopathological characteristics with NFIB expression.

Variable	n	NFIB		Adjusted P value	
		Low expression	High expression		
Age					
<60	149	72	77	0.526	
>60	301	155	146		
Sex					
Male	275	139	136	0.598	
Female	175	84	91		
Tumor laterality					
Left	227	118	109	0.368	
Right	220	105	115		
Histologic grade					
G1+G2	280	157	123	0.008 ∗	
G3+G4	113	48	65		
Pathologic stage					
I+II	255	141	114	0.011 ∗	
III+IV	146	63	83		
Tumor stage					
T1+T2	247	134	113	0.035 ∗	
T3+T4	168	75	93		
Lymph node status					
N0	318	171	147	<0.001 ∗	
N1	126	44	82		
Metastasis status					
M0	140	81	59	0.019 ∗	
M1	280	128	152		

NFIB promotes the proliferation and migration of KIRC cells.

To investigate the biological roles of NFIB, the mRNA and protein expression of NFIB was upregulated by transfection with plasmids containing the NFIB sequence (NFIB vector) compared to the vector control (NC vector) in the LoMet-ccRCC and 786-0 cell lines (Fig. 2A). The results showed that NFIB overexpression significantly increased the proliferation of the LoMet-ccRCC and 786-0 cell lines (Fig. 2B). We further used colony formation experiments to demonstrate that the overexpression of NFIB promoted the number of tumor cell colonies formed in the LoMet-ccRCC and 786-0 cell lines (Fig. 2C and 2D). In addition, the migration ability of LoMet-ccRCC and 786-0 cells was significantly increased after NFIB overexpression compared with NC vector transfection in transwell, invasion and wound-healing assays (Figs. 2E–2J). Additionally, siRNAs against human NFIB (NFIB-si) were applied to knockdown the expression of NFIB in KIRC cells (Fig. 3A). The knockdown of NFIB significantly repressed the proliferation and colony formation of LoMet-ccRCC and 786-0 cells (Figs. 3B–3E). Furthermore, transwell, invasion and wound-healing assays showed that NFIB depletion caused an obvious suppression of the migration ability of KIRC cells (Figs. 3F–3M). Therefore, the above results suggest that NFIB might be a promoter of KIRC tumorigenesis.

Figure 2 The high-expression of NFIB promotes the proliferation and migration of KIRC cells.

(A) The promoting efficiency of NFIB vector was evaluated by RT-qPCR and western blotting in LoMet-ccRCC and 786-0 cells respectively. (B) LoMet-ccRCC and 786-0 cells were transfected with NFIB vector or negative control (NC) vector for 24 h, and the proliferative ability was assessed by MTT assay over a 5-day period. (C–D) Colony formation assay (1×) was performed in LoMet-ccRCC and 786-0 cells transfected with NFIB vector or NC vector for 15 days. Migration ability was assessed in LoMet-ccRCC and 786-0 cells transfected with NFIB vector or NC vector by (E–F) transwell assay, (G–H) invasion assay, and (I–J) wound healing assay (200×), Scale bars = 100 µm. All data were presented as means ± SD of three independent experiments. Values are significant at * P < 0.05, as demonstrated by paired Student’s t test.

Figure 3 The low-expression of NFIB inhibits the proliferation and migration of KIRC cells.

(A) The inhibiting efficiency of NFIB siRNA was evaluated by RT-qPCR and western blotting in LoMet-ccRCC and 786-0 cells respectively. (B) LoMet-ccRCC and (C) 786-0 cells were transfected with NFIB siRNA or negative control (NC) siRNA for 24 h, and the proliferative ability was assessed by MTT assay over a 5-day period. (D–E) Colony formation assay was performed in LoMet-ccRCC and 786-0 cells transfected with NFIB siRNA or NC siRNA for 15 days. Migration ability was assessed in LoMet-ccRCC and 786-0 cells transfected with NFIB siRNA or NC siRNA by (F–G) transwell assay, (H–J) invasion assay, and (K-M) wound healing assay (200×), Scale bars = 100 µm. All data were presented as means ± SD of three independent experiments. Values are significant at * P < 0.05, as demonstrated by paired Student’s t test.

PINK1 is a potentially target of NFIB

To investigate the target genes of NFIB, we first performed functional enrichment analysis of the differentially expressed genes between KIRC and adjacent tissues obtained from three datasets (Fig. 1A and Table S5) using the DAVID online analysis website (https://david.ncifcrf.gov) (Jiao et al., 2012). These genes are significantly enriched in multiple KEGG pathways involved in tumorigenesis (Fig. 4A), such as pathways in cancer (P = 0.001) and MAPK signaling pathway (P = 0.002), associated with tumor cell proliferation, ECM-receptor interaction (P < 0.001) and the calcium signaling pathway (P < 0.001), associated with drug transport and metabolism in tumor cells. Given that NFIB is a transcription factor, we next investigated the possible regulation of NFIB by analyzing the gene promoters enriched in these pathways. The open database JASPAR (http://jaspar.genereg.net) and PROMO (http://alggen.lsi.upc.es/cgi-bin/promo_v3/promo/promoinit.cgi?dirDB=TF_8.3) were used to predicate the potential NFIB sites on the PINK1 gene promoter. Fortunately, we found three potential binding sites on PINK1 gene promoter for NFIB (Fig. 4B) (Khan et al., 2018; Messeguer et al., 2002), and bioinformatics analysis revealed that PINK1 was enriched in focal adhesion and pathways in cancer (Table S5). Hence, we hypothesized that PINK1 is one of the targets by which NFIB promotes KIRC progression and metastasis.

Figure 4 PINK1 was the pivotal downstream of NFIB to promote the progress of KIRC.

(A) Important KEGG pathways of differentially expressed genes in three datasets. (B) Potential binding sites of NFIB on the PINK1 gene promoter. After being transfected with NFIB vector or NC vector, respectively, the expression of PINK1 mRNA and protein in LoMet-ccRCC and 786-0 cells were analyzed by (C) qRT-PCR and (D) western blot assay respectively. After being transfected with NFIB siRNA or NC siRNA, respectively, the expression of PINK1 mRNA and protein in LoMet-ccRCC and 786-0 cells were analyzed by (E) qRT-PCR and (F) western blot assay respectively. (G) CHIP experiment was performed by using the NFIB and IgG antibodies to probe LoMet-ccRCC cells extracts, and the level of the co-precipitated RNAs were determined by using qRT-PCR. The data C, E and G were presented as means ± SD of three independent experiments. Values are significant at *P < 0.05, as demonstrated by paired Student’s t test.

NFIB regulates PINK1 expression

To further verify the role of NFIB, the KIRC cell lines LoMet-ccRCC and 786-0 were infected with NFIB vector or NC vector. Using qRT-PCR analysis, we found that NFIB overexpression significantly induced an increase in the mRNA expression level of PINK1 and that PINK1 expression was not induced by the NC vector (Fig. 4C). Furthermore, we used western blotting to detect the effect of NFIB overexpression on the expression of PINK1 protein. As expected, the protein level of PINK1 was significantly increased by NFIB overexpression compared with NC vector transfection (Fig. 4D). On the other hand, we observed the opposite results in KIRC cell lines after NFIB knockdown. qRT-PCR and western blotting showed that the knockdown of NFIB downregulated the expression levels of PINK1 mRNA and protein (Fig. 4E and 4F), respectively. As shown in Fig. 4B, three putative NFIB transcriptional binding sites in the PINK1 gene promoter region were found via bioinformatics analysis. Next, we performed a ChIP assay in LoMet-ccRCC and 786-0 cells using NFIB-specific antibodies. As shown in Fig. 4G, based on the qRT-PCR reads and the ChIP analysis results, we found that the NFIB-bound complex was remarkably enriched in PINK1 promoter site 2 but not site 1 or site 3. These results strongly implied that NFIB promotes KIRC metastasis and progression by regulating the promoter of PINK1.

PINK1 is the critical downstream of NFIB in KIRC

To confirm that PINK1 is a key factor through which NFIB promotes KIRC progression, we first conducted a rescue experiment. The cotransfection of PINK1 siRNA with NFIB vector dramatically decreased NFIB-upregulated PINK1 mRNA expression (Fig. 5A and 5B), and western blotting also obtained consistent results, and the promotion of PINK1 protein expression by NFIB was inhibited by PINK1 siRNA (Fig. 5C). We further examined the effect of PINK1 siRNA on the cell proliferation and progression induced by the overexpression of NFIB using LoMet-ccRCC cells. As expected, under the action of PINK1 siRNA, the increased proliferation and migration abilities of the LoMet-ccRCC cell line after NFIB overexpression were inhibited again (Figs. 5D–5L). On the contrary, we used CRISPR-Cas9 to knockout NFIB and express exogenous PINK1 under a constitutive expression promoter to observe the effects of KIRC cell migration, invasion and reproduction. We found that NFIB knockout resulted in a significantly down-regulation of PINK1 (Fig. 6A), and the proliferation and migration ability of KIRC cells was significantly reduced (Figs. 6B–6J). Then, after the exogenous overexpression of PINK1, the proliferation and migration ability of KIRC cells were significantly restored (Figs. 6A–6J). Therefore, PINK1 was the key downstream of abnormal expression of NFIB leading to abnormal biological behavior of KIRC cells. Taken together, these results suggested that PINK1 is a critical target of NFIB and that PINK1 exerts tumor-promoting roles in KIRC.

Figure 5 PINK1 knockdown inhibits the carcinogenic effects of NFIB.

After co-transfection with NFIB vector and PINK1 siRNA or NC siRNA respectively, the expression of PINK1 mRNA and protein in LoMet-ccRCC and 786-0 cells were measured by (A–B) qRT-PCR and (C) western blot assay respectively. The proliferative capacity in LoMet-ccRCC cells were analyzed by (D) MTT and (E–F) colony formation assay (1×). The migration ability in LoMet-ccRCC cells were analyzed by (G–H) transwell assay, (I–J) invasion assay and (K–L) wound healing assay (200×), Scale bars = 100 µm. All data were presented as means ± SD of three independent experiments. Values are significant at *P < 0.05, as demonstrated by paired Student’s t test.

Figure 6 PINK1 is the critical downstream of NFIB in KIRC.

After co-transfection with NFIB sgRNA and PINK1 vector or NC vector respectively, the expression of PINK1 protein in LoMet-ccRCC and 786-0 cells were measured by (A) western blot assay. The proliferative capacity in 786-0 cells were analyzed by (B) MTT and (C–D) colony formation assay (1×). The migration ability in 786-0 cells were analyzed by (E–F) transwell assay, (G–H) invasion assay and (I–J) wound healing assay (200×), Scale bars = 100 µm. All data were presented as means ± SD of three independent experiments. Values are significant at *P < 0.05, as demonstrated by paired Student’s t test.

Discussion

NFIB has been reported to play pivotal roles in the tumorigenesis of cancers, including gastric cancer, breast cancer, melanoma, and nonsmall cell lung cancer (Campbell et al., 2018; Fane et al., 2017; Wu et al., 2017; Wu et al., 2018). However, at present, there was no report about the relationship between NFIB mutation and tumor, whether the expression changes of NFIB are related to specific pathological subtypes. Moreover, the functions and mechanisms of NFIB in KIRC tumorigenesis are far from defined. Therefore, it is necessary to further study the role of NFIB in the development of KIRC and the corresponding molecular mechanisms to help to better understand the mechanism of tumorigenesis and to provide novel and valuable targets for drug therapy. A significant new finding in this study is that NFIB plays a role in the proliferation and metastasis of KIRC by regulating PINK1 expression, and ChIP experiments confirmed that NFIB acts primarily through binding to the PINK1 promoter.

We first analyzed TCGA and GEO datasets and found that NFIB is one of the significantly increased genes in KIRC. Further analysis of the correlation between NFIB and KIRC clinical pathological parameters revealed that the high expression of NFIB was significantly associated with KIRC tumors with poor grades, significant metastasis, and worse prognosis. In view of these results, we consider it necessary to further investigate the role of NFIB in KIRC tumor growth and metastasis. With the overexpression of NFIB, the proliferation and migration ability of KIRC tumor cells was significantly improved. Conversely, the knockdown of NFIB significantly reduced the proliferation and metastasis ability of KIRC cells. These results indicate that NFIB may support the progression of KIRC.

To further explore the mechanisms by which NFIB promotes KIRC progression, we performed functional enrichment analysis on DEGs in KIRC in the databases. As shown in Fig. 2A, we found that multiple pathways are enriched in cell–cell interactions, including cell adhesion molecules, focal adhesion and tight junctions. Moreover, enrichment analysis indicated that multiple pathways are involved in the malignant proliferation of KIRC tumor cells, including the MAPK signaling pathway and pathways in cancer. In addition, the transport functions of the cell membrane play a vital role in the progression of KIRC, including the calcium signaling pathway. Since NFIB is a nuclear transcription factor that plays a major role in the transcriptional regulation of target genes, we wanted to explore the downstream targets of NFIB by analyzing the promoters of genes enriched in the abovementioned pathways. We found that the expression level of PINK1 were indeed altered by the regulation of NFIB expression.

PINK1 encodes a serine/threonine protein kinase that is primarily localized in the mitochondria (Nguyen, Padman & Lazarou, 2016). PINK1 was originally found to be upregulated by the overexpression of the tumor suppressor PTEN in HeLa cells, which is thought to protect cells from stress-induced mitochondrial dysfunction (Di Rita et al., 2018). In addition, mutations in PINK1 cause autosomal recessive early-onset Parkinson’s disease (Ren et al., 2019). Recent studies have reported the role and mechanism of PINK1 in tumor proliferation, metastasis, apoptosis and tumor resistance (Liu et al., 2018; Zhang et al., 2017). PINK1 interacts with the oncogenic PI3K/Akt/mTOR signaling axis at multiple levels and controls the regulation of cancer survival and growth (Wang et al., 2019). Moreover, studies have indicated that PINK1 has cytoprotective and chemoresistant functions in breast cancer and that PINK1 can be used as a target for breast cancer treatment (Li et al., 2017). Related studies in lung cancer have also shown that the silencing of PINK1 inhibits the migration and invasion of lung cancer cells and that the inhibition of PINK1 enhances the apoptosis rate of cancer cells (Liu et al., 2018). In summary, PINK1 has a strong connection with the malignant development of tumors. Moreover, we used an online analysis site to predict possible binding sites of NFIB on the PINK1 gene promoter (Khan et al., 2018; Messeguer et al., 2002). Furthermore, ChIP experiments further confirmed that “GGGATTCACC” on the PINK1 gene promoter sequence is a binding site of NFIB, which further suggests that PINK1 is directly regulated by NFIB. To demonstrate the critical role of PINK1 in the promotion of KIRC mediated by NFIB, it was necessary to conduct a “rescue” experiment. As expected, when the NFIB vector and the small interfering RNA against PINK1 were simultaneously transfected into the LoMet-ccRCC cell lines, the NFIB-induced promotion of cell proliferation and metastasis was significantly inhibited.

Although our studies needs to be further explored in vivo, and the relationship between NFIB and classic oncogenic signaling pathways needs further discussion. However, in conclusion, our study confirmed multiple pathways involved in KIRC progression and further demonstrated the role of NFIB in promoting KIRC progression and metastasis through the transcriptional regulation of PINK1 expression. Moreover, our data analysis found that NFIB expression levels are related to multiple clinical pathological parameters of KIRC and to patient prognosis; thus, NFIB could be used as a potential biomarker to clinically evaluate the severity of KIRC in patients.

Conclusions

In conclusion, our study demonstrated the role of NFIB in promoting KIRC progression and metastasis through the transcriptional regulation of PINK1 expression. Moreover, we suggested that NFIB could be used as a potential biomarker to clinically evaluate the severity of KIRC in patients.

Supplemental Information

Supplemental Information 1 Expression matrix of RNA sequence from TCGA

Click here for additional data file.

Supplemental Information 2 Expression matrix of RNA sequence from GSE83999

Click here for additional data file.

Supplemental Information 3 Expression matrix of RNA sequence from GSE53757.

Click here for additional data file.

Supplemental Information 4 Clinical characteristics of KIRC patients

Click here for additional data file.

Supplemental Information 5 KEGG analysis of the differentially expressed genes

Click here for additional data file.

Supplemental Information 6 Raw data of qPCR

Click here for additional data file.

Supplemental Information 7 Full-length uncropped blots

Click here for additional data file.

We sincerely appreciate our team members for providing statistical instructions.

Additional Information and Declarations

Competing Interests

Author Contributions

Human Ethics

Data Availability

The authors declare there are no competing interests.

Ninghua Wang conceived and designed the experiments, analyzed the data, authored or reviewed drafts of the paper, and approved the final draft.

Jing Yuan conceived and designed the experiments, analyzed the data, prepared figures and/or tables, authored or reviewed drafts of the paper, and approved the final draft.

Fei Liu performed the experiments, prepared figures and/or tables, and approved the final draft.

Jun Wei performed the experiments, authored or reviewed drafts of the paper, and approved the final draft.

Yu Liu performed the experiments, analyzed the data, prepared figures and/or tables, and approved the final draft.

Mei Xue and Rui Dong analyzed the data, authored or reviewed drafts of the paper, and approved the final draft.

The following information was supplied relating to ethical approvals (i.e., approving body and any reference numbers):

The study was approved by the Ethics Committee of the Academic Medical Center of Wuhan University of Science and Technology (Ethical Application Ref: 2020-IEC1453).

The following information was supplied regarding data availability:

Raw measurements are available in the Supplemental Files.

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
