# Peer review of "NFIB promotes the migration and progression of kidney renal clear cell carcinoma by regulating PINK1 transcription"

_PeerJ, doi:10.7717/peerj.10848_

## Round 0.1 · original submission · Major Revisions

Please address all comments by the reviewers.

Please address the following:

1) In vitro migration does not necessarily mimic in vivo metastasis. Please perform an invasion assay at the minimum.

2) Please knockout PINK1 (using CRISPR-Cas9) in the NFIB silenced cells that are being used in the study and express exogenous PINK1 under a constitutively expressing promoter looking at the effect of KIRC cell migration, invasion, and proliferation. If PINK1 overexpression in NFIB silenced cells do not rescue the phenotype, it would possibly suggest an alternate pathway mediated by NFIB in KIRC cell migration., invasion, and proliferation.

3) Please give a detailed methodology for the RNA Seq. Please mention the basis for selecting PINK1 for further analysis over other differentially expressed genes. Why was the study further pursued in cell lines? Please mention the limitations of using cell line analysis over an in vivo system.

Reviewer 1 ·

Basic reporting

1) The abstract contains acronyms. Please define before using acronyms.
2) Introduction (line 33: use more recent statistics)
3) Page 2, Line 40, Kim et al 2011 is an incorrect citation. This paper reports 29.1% deaths and deals with RCC, not KIRC
4) Page 2, Line 44, Hanahan 2008 doesn't mention KIRC
5) Metastasis is in an organism. The authors do not demonstrate metastasis in mice etc. Hence, the change of title is required from metastasis to migration.

Experimental design

Materials and methods:
1) Under transfections, please provide the siRNA targetting sequence for NFIB and PINK1
2) Please include treatments/experimental groups under colony-forming assay
3) Please write the full form of IHC and ChIP on page 4 line 137 and 145 respectively
4) Please include scale bars in Fig 2D,E, I, J

Validity of the findings

This study demonstrates the role of NFIB in promoting KIRC migration and proliferation through the transcriptional activation of PINK1. ChIP experiments confirmed the binding sequence on the PINK1 gene promoter. This study is important and of significance since this molecular interaction could result in the activation of pro-oncogenic pathways in cancer cells.

Additional comments

I propose the following experiments for this paper to be accepted for publication.
1) In vitro migration does not necessarily mimic in vivo metastasis. I suggest the authors perform an invasion assay at the minimum.
2) I suggest the authors knockout PINK1 (using CRISPR-Cas9) in the NFIB silenced cells that are being used in the study and express exogenous PINK1 under a constitutively expressing promoter looking at the effect of KIRC cell migration, invasion, and proliferation. If PINK1 overexpression in NFIB silenced cells do not rescue the phenotype, it would possibly suggest an alternate pathway mediated by NFIB in KIRC cell migration., invasion, and proliferation.

Reviewer 2 ·

Basic reporting

Overall, the writing is clear. Certain sections have been flagged in my comments below.

Experimental design

No comment.

Validity of the findings

Some conclusions have been overstated and in some cases, the effect size is exaggerated in the text. These sections have been flagged in my comments below.

Additional comments

In this manuscript, Wang and colleagues find expression of a DNA-binding protein, NFIB, to be upregulated in kidney renal clear cell carcinoma (KIRC). They find NFIB to bind the promoter sequence of PINK1 to regulate its transcription. While the authors present convincing data for these conclusions, some of their other inferences are premature. There is no clear data showing that the tumorigenic functions of NFIB are in fact mediated by PINK1. The recue experiments conducted by the authors are poorly designed and do not address this question. Also, the authors claim NFIB/ PINK1 to be regulators of metastasis with no supporting data. These areas must be address in the form of simple new experiments and re-writing of the text.

Major comments:
1. The rationale behind the design of rescue experiments conducted by authors in Figure 4 is unclear to me. Why are PINK1 expression levels (Fig. 4a, b) measured when the cells are treated with a PINK1 siRNA? What is the conclusion drawn by these experiments?
2. The authors do not show effects of PINK1 overexpression or knockdown (in the absence of NFIB overexpression). These experiments are important to further implicate PINK1 in KIRC.
3. The authors show images from colony formation, invasion, and migration assays in Figure 2 and 4 to test tumorigenic properties of NFIB and PINK1. Yet, they provide no quantification for these assays. These end points of these assays can be easily quantified and must be included in the manuscript.
4. Western blots confirming the knockdown or overexpression of NFIB/ PINK1 must be included.
5. The images chosen for transwell invasion assay comparing NC and NFIB overexpression vectors in LoMet-ccRCC cell line do not seem to show appreciable differences. Similarly, the authors must comment in the results section that neither overexpression nor knockdown of NFIB seems to have an effect on the migration (wound healing assay) of 786-0 cells.
6. Throughout the entire manuscript (abstract through discussion), the authors state that both NFIB and PINK1 regulate KIRC metastasis. While the in vitro studies performed in this manuscript implicate these proteins in promoting aspects of tumorigenesis, their role in metastasis is unexplored in this study. The authors should rewrite portions of the text that overstate their conclusions.
7. What is the scoring criteria that authors use for their IHC data? There is no information provided in the methods section. Additionally, NFIB is a nuclear protein and it must be clarified if protein localization to the nucleus is taken into account for scoring purposes.
8. The authors should state the statistical test used for determining p-values in the figure legends.

Minor comments:
1. Line 40: Vague phrasing of statistic. Unclear what the 51% of cases corresponds to. The original statistic in Kim et al. 2011 is that the median survival of patients that die from renal cell carcinoma (29.1%) is 1.9 years.
2. Line 67” “In cell experiments” should be written as “in vitro” or “in cell lines”
3. Line 191: I recommend authors chose between GEPIA or KMPLOT plots. They show similar things from similar datasets and having two panels is redundant.
4. Line 209: It is not clear what comparison was made in obtaining the DE gene list. I presume it is tumor vs normal adjacent tissue in KIRC but this should be explicitly stated here.
5. Line 217: I presume the authors identified targets whose promoters have the NFIB binding consensus sequence. The analysis methodology must be described more clearly.
6. Line 269: “KIRC may facilitate the development of KIRC.” Two problems- the first KIRC is a typo and should be NFIB. Also, there is no evidence in this study that implicated NFIB in the development of KIRC, rather it supports tumor progression.

Reviewer 3 ·

Basic reporting

This manuscript identifies that NFIB promotes tumor proliferation and metastasis by up-regulating the transcriptional expression of PINK1. The topic of the manuscript is important as the identification and characterization of tumorigenesis and metastasis is key to understand cancer biology, which ultimately may improve diagnostics and therapeutics. The role of NFIB in tumor invasion and progression has been reported and studied in multiple cancer types and the authors report it for the first time in KIRC. The authors analyze gene expression data to identify differentially expressed genes and ultimately identify PINK1 as a NFIB target. Overall the manuscript is well structured and not many grammatical or syntax mistakes were found

Experimental design

The main issues in this paper is the lack of clear methods. It is not clear how the authors did the RNA-seq analysis to identify the differentially expressed genes. In addition, it is not clear how the authors decided to focus on PINK1 after the gene promoter analysis. The research question is well defined and it is well stated. It is also important to note that the authors should adjust their p-values throughout the manuscript.

Validity of the findings

The findings seem to be valid. However, clear methodology will help support the conclusions of the manuscript. It should also be noted that the results obtained in this paper for the role of NFIB/PINK1 is based on cell line data, which not always represents exactly what happens in tumorigenesis and cancer biology. Therefore, the authors should emphasize the limitations of using cell lines in the study. It will also enrich the manuscript if the authors analyze the mutational landscape of these tumors, as PINK1 interacts with the oncogenic PI3K/Akt/mTOR and these genes could be altered in some tumors. Could the NFIB or PINK1 over-expression be present in a subset of samples with a specific type of alterations? Could somatic copy number variation play a role in NFIB/PINK1 over-expression? Do the samples analyzed have structural alterations in these genomic regions?

Additional comments

- The RNA-seq analysis is not clearly explained in methods as it only references a previous publication. It should be stated what data was acquired (fastq, TPM expression?) for each of the studies used. In addition, was all the expression data processed uniformly? What software, version and parameters were used to obtain the differentially expressed genes?
- What are the units shown on Figure 1B?
- Is the y-axis correct in Figure 1C? I believe adjusted p-values should be used
- Are the * in Figure 1D adjusted p-values? Is it also over-expressed in other renal cancer cell lines or is this a KIRK specific?
- Table 1 should use adjusted p-values and results should be reported accordingly
- Show hazard ratio and its 95% confidence interval, and the N for low and high NFIB expression. Shouldn’t the authors adjust their p-value in between 1H-J. Color scale should be consistent between the three plots. In addition, tables with demographic and clinical data should be provided for the three cases, to show similar patient populations on low vs high expression. Are there other differentially expressed genes also associated with prognosis for KIRK? Is NFIB one of the top prognostic related genes?
- Show median overall survival for high vs low NFIB expression
- Is NFIB an essential gene? Is it associated with synthetic lethality based on CRISPR screens such as DepMap or ProjectScore
- Line 218 How was the potential TF binding site predicted? Was this analysis only done for PINK1 or all differentially expressed genes? It is not clear how the authors determine to only study PINK1 based on NFIB analysis.

---

## Round 0.2 · Minor Revisions

Thank you for addressing all concerns by the reviewers. Please address comments by Reviewer 2.

Reviewer 1 ·

Basic reporting

no comment

Experimental design

no comment

Validity of the findings

no comment

Reviewer 2 ·

Basic reporting

Abstract:
1. Line 15: explicitly list the databases used for the bioinformatic analysis.
2. Line 15: The bioinformatic analysis provided rationale for the authors to investigate NFIB. Yet, they say that they use this analysis to "confirm" it. Rephrasing is recommended strongly.
3. Line 23: Experiment"s"
4. Line 25: The authors do not provide any data to implicate NFIB/ PINK1 in the development of KIRC. This must be rewritten as KIRC tumor progression.

Main text:
1. Line 203: "worse" prognosis
2. As I mentioned in my original comments, it is unclear if the authors show protein level changes in the expression of NFIB and PINK1. In their response they said that this data is included. I presume they are referring to panel A in these figures. It is unclear if this is WB or qPCR data. This should be clarified in the main text and also the figure legends.
3. Line 235-236: no data for "associated with tumor cell metastasis". This must be deleted for accuracy.
4. Line 238: Delete "nuclear". All transcription factors are nuclear proteins.
5. Line 260: Tumor progression, not metastasis.
6. Line 265: Tumor progression, not metastasis.
7. Recommend moving lines 278-282 (and corresponding figure panel) to Fig 4. It flows a lot better in the manuscript.

Figure legends:
1. The figure legends have not been updated since the revision. This is true for all the figures.
2. Legend for Figure 1: Data in G can not be from three independent experiments. I presume each dot is a separate tissue specimen.

Figures:
1. Fig 1H: Why does the Kaplan Meyer plot say RAB17 on top? These curves should be for NFIB, not RAB17.
2. Fig 2F, 3G, 5H, 6F: Why is the axis labelled "metastasis number." The authors are counting the number of cells invaded in a transwell assay.
3. Fig 2J, 3M, 5L, 6J: Axis label should be "relative" wound size
4. Fig 5K, 6I: The authors should flip the orientation of the images to match those of similar assays in earlier figures. This is important so the differences between the different treatment groups can be readily compared.
5. All figures need scale bars.
6. General stylistic comment: The authors frequently use circles/ squares in dot plots/ curves to differentiate 2 conditions. I highly recommend the use of different colors so that these figures can be interpreted more easily.

Experimental design

No additional comments.

Validity of the findings

No additional comments. Few instances where authors overstate conclusions have been flagged in the "basic reporting" comment section.

Additional comments

No additional comments.

External reviews were received for this submission. These reviews were used by the Editor when they made their decision, and can be downloaded below.

---

## Round 0.3 · accepted · Accept

Thanks for addressing all reviewers' comments. Please incorporate few minor comments suggested by Reviewer 3.

Reviewer 2 ·

Basic reporting

No comments

Experimental design

No comments

Validity of the findings

No comments

Additional comments

No comments

Reviewer 3 ·

Basic reporting

The plots of 1H and 1I should be in the same format as they represent the same kind of data. In addition, 1I y axis says percent when the values shown are fractions

Line 263. How do the results “strongly imply strongly implied that NFIB promotes KIRC metastasis and progression by regulating the promoter of PINK1”? The authors should tone down and use terms like “suggests”

Experimental design

How did the authors perform batch correction for the RNA seq analysis?


How was the promoter transcription factor binding site analysis done? How was a promoter defined? For only the reference transcript or isoforms were also included? Was PINK1 the only gene showing NFIB binding sites? What is the “bioinformatics analysis” that showed the pathways enriched for PINK1? All of these should be in the methods and results section

Authors should also note that regulation goes beyond promoters, for example enhancers have not been analyzed

Validity of the findings

For figure 1B, I would suggest the authors use a color annotation to show the dataset of origin for each sample. This will also show if there is a batch effect if samples from the same dataset cluster together

Additional comments

Overall, the manuscript has improved. I would recommend that the authors improve the methods section to be more precise

External reviews were received for this submission. These reviews were used by the Editor when they made their decision, and can be downloaded below.